# Ordered Domain (Raft) Formation in Asymmetric Vesicles and Its Induction upon Loss of Lipid Asymmetry in Artificial and Natural Membranes

**DOI:** 10.3390/membranes12090870

**Published:** 2022-09-09

**Authors:** Erwin London

**Affiliations:** Department of Biochemistry and Cell Biology, Stony Brook University, Stony Brook, NY 11794-5215, USA; erwin.london@stonybrook.edu

**Keywords:** lipid asymmetry, phospholipids, cholesterol, plasma membranes

## Abstract

Lipid asymmetry, the difference in the lipid composition in the inner and outer lipid monolayers (leaflets) of a membrane, is an important feature of eukaryotic plasma membranes. Investigation of the biophysical consequences of lipid asymmetry has been aided by advances in the ability to prepare artificial asymmetric membranes, especially by use of cyclodextrin-catalyzed lipid exchange. This review summarizes recent studies with artificial asymmetric membranes which have identified conditions in which asymmetry can induce or suppress the ability of membranes to form ordered domains (rafts). A consequence of the latter effect is that, under some conditions, a loss of asymmetry can induce ordered domain formation. An analogous study in plasma membrane vesicles has demonstrated that asymmetry can also suppress domain formation in natural membranes. Thus, it is possible that a loss of asymmetry can induce domain formation in vivo.

## 1. Membrane Lipid Asymmetry and Loss of Asymmetry

Many natural membranes have strong lipid asymmetry in which the headgroup and acyl chain compositions are different in the inner and outer lipid monolayers (leaflets) that form the lipid bilayer [1]. The most familiar case is that of erythrocytes, in which the presence of a single cell membrane allows relatively facile measurement of lipid asymmetry. However, cultured mammalian cells, and other eukaryotes, have plasma membranes with asymmetry similar to that of erythrocytes [2]. It has been shown repeatedly that in mammalian erythrocytes, sphingomyelin (SM), and to a lesser extent phosphatidylcholine (PC), are enriched in the outer leaflet, while phosphatidylethanolamine (PE) and phosphatidylserine (PS) are concentrated in the inner leaflet. Glycosphingolipids are generally outer leaflet lipids while phosphatidylinositol (PI) and its phosphorylated variants appear to be mainly on the inner leaflet. Cholesterol is located in both leaflets, but the amounts in each leaflet are still controversial. Classical and recent studies of asymmetry differ somewhat on the degree of asymmetry for several lipids, and it is not clear if this represents a difference in experimental protocols or experimental conditions [1,3].

In addition to random transverse diffusion of lipids across the bilayer of a membrane (see below), asymmetry is controlled by the sidedness of lipid biosynthesis and breakdown, plus the presence of lipid translocases, which include proteins that transport lipids inwardly (i.e., towards the cytoplasm) (flippases) or outwardly (floppases), and scramblases, which facilitate random diffusion of lipids in both directions [4].

Loss of lipid asymmetry has biological impacts. Loss of PS asymmetry is associated with apoptosis and triggering of blood clotting, in which the appearance of PS in the outer leaflet promotes interactions with phagocytic cells and blood clotting factors, respectively [4]. It has more recently also become appreciated that transient loss of asymmetry is associated with signal transduction events [5]. (It is noteworthy that at least some lipid scramblases (e.g., TMEM16F) are activated by cytosolic Ca^2+^, which increases in concentration during some types of signal transduction [4].) Cancer cells can also have decreased asymmetry [4]. In addition, viruses that bud off of cells can have membranes with reduced PS asymmetry and have enhanced infection via apoptotic mimicry, in which outer leaflet PS in the viral membrane induces viral uptake by phagocytic cells [6,7]. Loss of asymmetry for other lipids may also have biological consequences. For example, loss of PE asymmetry may contribute to processes enhanced by loss of PS asymmetry [8]. In addition, a recent study has identified effects of lost SM asymmetry on membrane repair processes [9]. Thus, the biological impacts of changes in asymmetry are of wide importance.

## 2. Preparation of Asymmetric Lipid Vesicles with Cyclodextrins

Despite the above discussion, our knowledge of how asymmetry and changes in asymmetry control membrane structure and function remains rudimentary. This is largely due to the difficulty of measuring membrane asymmetry for lipids other than PS, and limitations in experimental control of membrane lipid composition and asymmetry. Lack of good methods for preparation of asymmetric artificial membranes in which the biophysical effects of membrane asymmetry can be studied has been one major stumbling block. Advances in the preparation of artificial and natural membranes with controlled asymmetry are starting to change this situation [10,11]. For details on various methods, the interested reader is directed to some recent reviews [11,12,13,14]. It is now possible to prepare highly stable asymmetric lipid vesicles that imitate the asymmetry in natural membranes using a number of methods, with cyclodextrin-catalyzed lipid exchange having wide applicability to small, large and giant vesicles [11]. Cyclodextrins are cyclic glucose oligomers that can accommodate hydrophobic molecules in their central cavity. In the cyclodextrin-based method, the outer leaflet lipids of two populations of lipid vesicles with different lipid compositions are exchanged by incubating them in the presence of membrane impermeable, water soluble cyclodextrins that can bind membrane lipids. Since the cyclodextrins used have only a moderate lipid binding strength, they bind and dissociate lipids so as to equilibrate the outer leaflet compositions of the lipid vesicles. The vesicles with the desired composition can then be isolated by centrifugation or filtration [15,16,17]. Beta class cyclodextrins are often used to alter sterol levels in membranes. Alpha class cyclodextrins can bind lipid acyl chains, but do not accommodate cholesterol, allowing exchange of phospholipids without perturbing membrane sterol content [18].

Importantly, the cyclodextrin-catalyzed exchange method has been extended to mammalian cells. Efficient lipid exchange can be achieved in erythrocytes and cultured cells [2,19]. This opens up the possibility of defining how lipid composition and lipid asymmetry influence membrane function in living cells [20,21]. In addition, by identifying the amount of each lipid removed from the outer leaflet of the plasma membrane upon exchange, the method should be useful for assay of lipid asymmetry under different conditions [2].

Another method proving highly useful for preparation of asymmetric giant artificial lipid vesicles involves hemifusion between giant vesicles and a lipid covered surface [10].

## 3. Stability of Lipid Asymmetry in Artificial Lipid Vesicles

An important parameter is the stability of lipid asymmetry. When lipids undergo transverse diffusion across the bilayer (i.e., flip between leaflets) asymmetry decays. In cell membranes the stability of asymmetry is often controlled by proteins. However, spontaneous lipid flip across the bilayer also occurs. In both symmetric and asymmetric membranes the spontaneous loss of lipid asymmetry in flip across the bilayer unaided by protein is generally very slow for the most commonly occurring phosphatidylcholines or sphingomyelins, on the order of hours to days in the liquid disordered state, and even slower in the gel state (see below for what distinguishes these physical states) [22]. In contrast, lipid flip can be fast (minutes) in membranes composed of phosphatidylcholines with two polyunsaturated chains [23]. (However, such lipids are rare in nature.) Lipid polar headgroup can also affect lipid flip. In asymmetric vesicles containing SM in one leaflet and phosphatidylglycerol (PG), PI or cardiolipin (CL) in the opposite leaflet, SM flip (and thus that of the other lipids) seemed to occur in the range of hours, while asymmetric vesicles containing various mixtures of SM, PC, PE and PS, exhibited much slower flip [24], even when a moderate level of PG, PI or CL was also present. A high level of cholesterol does not result in fast phospholipid flip [18], even though cholesterol flips rapidly under most conditions [25]. It should be noted that lipid flip properties different from those in lipid vesicles have been reported in some supported bilayers [26], perhaps due to rapid flip at the site of bilayer defects [22].

The consequence of slow lipid flip is that it is easy to prepare asymmetric vesicles with lipid compositions that maintain lipid asymmetry for prolonged periods, including cases in which lipid composition mimics that in natural membranes, and this allows a wide range of experiments to be carried out.

However, it should be noted that in some (but not all) cases, inclusion and/or addition of membrane-inserting peptides to lipid bilayers can greatly increase the rate of flip-flop [27,28,29,30], and thus reduce asymmetry.

## 4. Membrane Physical State and Domain Formation

The formation of membrane domains in different physical states is a key membrane property believed to be of biological importance. It has been widely studied using lipid vesicles and has been a major subject of studies using asymmetric vesicles. The phospholipid/sphingolipid bilayer of a membrane often exists in one of three physical states. The most common is the liquid disordered (Ld) state, which is characterized by fast lateral diffusion of lipids and lipids having disordered acyl chains. Lipids with at least one of their hydrocarbon (acyl) chains having a cis double bond (i.e., being unsaturated) tend to form bilayers in this state at physiological temperatures. At low temperatures, or for lipids with two hydrocarbon chains lacking cis double bonds (e.g., lipids with long saturated acyl chains), the solid-like, highly ordered gel state (G) forms at both low and higher temperatures. In the gel state, lipid hydrocarbon chains are tightly packed against each other and lateral diffusion is very slow. It should be noted that when temperature is varied a lipid bilayer can transition between liquid disordered and gel states at a characteristic transition (melting) temperature (Tm). As the discussion above indicates, lipids with at least one of their acyl chains being cis unsaturated tend to have low Tm values, while lipids with only saturated acyl chains (often SM or a saturated PC) tend to have high Tm values [31].

In membranes with saturated phospholipids (or sphingolipids) and sterol (in mammals, cholesterol), the liquid ordered (Lo) state can form. The liquid ordered state is characterized by a high level of order but relatively fast lateral diffusion. Recent studies indicate that the liquid ordered state is composed of gel/solid-like phospholipid or sphingolipid subdomains laterally separated by sterol-rich layers [32,33]. An analogy to the liquid ordered state would be a cluster of small icebergs separated by a thin layer of liquid so that they can laterally diffuse rapidly within the cluster relative to each other despite having a solid internal state.

A consequence of basic physical chemistry principles is that for some lipid compositions in membranes composed of a mixture of lipids with different tendencies to form different physical states, two or more of these physical states will co-exist, forming lipid domains with different physical states and lipid compositions [34]. The formation of lipid domains with different physical states in natural membranes is of biological interest for two reasons. First, preferential association of proteins with one type of domain can enhance interactions between proteins that prefer the same type of domains and inhibit interactions between proteins that prefer to localize in different domains. Second, the difference in lipid conformation in different physical states, which, for example, results in a thicker bilayer width in ordered states relative to the Ld state, can influence membrane protein conformation, and thus biological activity.

## 5. Range of Domain Properties in Asymmetric Membranes

How lipid composition controls membrane physical state and domain formation has been studied extensively in symmetric artificial membranes. Far fewer studies have been carried out in asymmetric artificial membranes, but those already carried out have provided a number of novel insights. One is that asymmetry can have a wide variety of impacts upon the physical properties of each leaflet when only one leaflet contains lipids that would spontaneously form ordered domains in a symmetric membrane [11]. For example, cases in which ordered domains in one leaflet induce ordered domains in the opposite leaflet, and cases in which ordered domain formation in a leaflet is suppressed by contact with a disordered leaflet, have both been observed (Figure 1). These phenomena occur when the physical properties of the two leaflets are coupled to each other. The leaflet whose physical state is the same in asymmetric vesicles and symmetric vesicles with the same lipid composition as that leaflet is said to be dominant. Which leaflet dominates can be temperature dependent, with the leaflet having lipids that can form ordered domains only dominant at lower temperatures (see below).

In the most common cases, the location of a domain in one leaflet matches the location of a domain having the same physical state in the opposite leaflet. In this case, the domains are said to be in register. However, there can be exceptions [35]. It should also be noted that in cases of strong coupling, the physical states in opposite leaflets would exactly match each other, but when coupling is weaker, partial coupling can occur so that the physical state of an induced ordered domain only changes part way towards matching that of the ordered domain in the opposite leaflet, or in which an ordered domain has looser than normal packing due to partial coupling to a disordered leaflet. It is also possible for the physical state of the lipids to be uncoupled, so that each leaflet is in a different physical state, the physical state expected for a symmetric vesicle of that leaflet composition.

## 6. Experimental Studies of Interleaflet Coupling in Asymmetric Vesicles without Sterol

In asymmetric small vesicles without cholesterol, exchanging in SM to replace almost the entire outer leaflet of vesicles containing dioleoyl PC (DOPC), which has two unsaturated acyl chains, or palmitoyl oleoyl PC (POPC), which has one unsaturated acyl chain, resulted in vesicles that, at ambient temperature, had a highly ordered outer leaflet, but with only a modest increase in inner leaflet order relative to that in symmetric DOPC or POPC vesicles [15,36], suggesting weak to moderate coupling of membrane order. Interestingly, outer leaflet Tm and order was similar to that for pure SM, indicating that the inner leaflet had little effect on outer leaflet properties. This low level of coupling did not seem to be a consequence of small vesicle size, i.e., vesicle curvature [16].

However, it was found in another study that the difference between inner and outer leaflet curvature could affect coupling greatly in asymmetric vesicles containing palmitoyl oleoyl PE (POPE) and POPC, with strong coupling of ordered state formation in both leaflets when the relatively high-Tm lipid POPE was enriched in the inner leaflet, but not when it was enriched in the outer leaflet [37].

In a study using large vesicles containing saturated dipalmitoyl PC (DPPC) plus POPC in one leaflet and POPC in the other, partial coupling was observed by neutron scattering, such that the DPPC rich ordered domain in the outer leaflet packed more loosely than it would in symmetric vesicles [17]. Ordered domain formation in the inner leaflet was not detected.

In diffusion studies in which SM was introduced into the outer leaflet of giant vesicles composed of various PCs, decreased lateral diffusion of lipids in each leaflet was detected after introduction of SM even though domains were too small to visualize. The extent of coupling of diffusion, as judged by the relative decrease in lateral diffusion in each leaflet after SM was introduced, depended on the choice of SM and PC. Upon introduction of brain SM into the outer leaflet, lateral diffusion of inner leaflet PCs with one saturated and one unsaturated acyl chain (e.g., POPC) was more strongly coupled to that of the outer leaflet than when the PC in the vesicles was DOPC, as judged by the decrease in inner leaflet lateral diffusion relative to the decrease in outer leaflet lateral diffusion after SM introduction. Coupling of diffusion in the two leaflets was also strong in DOPC containing membranes when milk SM was introduced into the outer leaflet instead of brain SM. Penetration of long milk SM acyl chains into the opposite leaflet (interdigitation) may have increased the extent of interleaflet coupling in those vesicles.

## 7. Experimental Studies in Asymmetric Vesicles in the Presence of Sterol: Induction of Ordered Domains Due to Interleaflet Coupling

Because domain formation in cells involves membranes containing cholesterol or an analogous sterol, the behavior of cholesterol-containing artificial membranes is of most interest. Natural sterols present at high concentrations, as in the plasma membrane, generally promote ordered domain formation. However, sterol effects depend strongly on their structure, and some sterols can even suppress ordered domain formation [38,39,40,41]. Interestingly, studies have shown that although the ability of sterols to promote ordered domain formation in symmetric and asymmetric vesicles is similar, they are not always identical. Epicholesterol is able to support ordered domain formation less well in asymmetric vesicles than symmetric vesicles while the opposite is true for desmosterol [42].

Early pioneering studies using unsupported or cushioned asymmetric planar bilayers containing cholesterol identified situations in which domains in one leaflet induced domains in the opposite leaflet, as well as those in which a lack of ordered domains in one leaflet appeared to suppress ordered domain formation in the opposite leaflet [43,44]. These microscopy-based studies had limitations due to residual solvent, ambiguities due to probe partitioning that can result in loss of contrast between domains, and/or the inability to distinguish a loss of domain formation from a transition of microscopic to submicroscopic domains. Use of asymmetric lipid vesicles can circumvent these issues by judicious probe choice, avoidance of solvents, and use of FRET or other methods to detect submicroscopic domains [17,45].

First, we consider cases in which ordered domains formed by SM or saturated PC in one leaflet (usually Lo domains in the presence of cholesterol) induce ordered domains in an opposite leaflet enriched in Ld-forming lipids, i.e., lipids that do not by themselves form ordered domains in symmetric membranes. In cholesterol-containing asymmetric giant vesicles prepared by cyclodextrin-catalyzed exchange, large (visible by light microscopy) outer leaflet ordered domains and interleaflet coupling-induced ordered domains in the inner leaflet were observed in vesicles composed of egg SM, DOPC and cholesterol in the outer leaflet and DOPC plus cholesterol in the inner leaflet [46]. It was found that the inner leaflet ordered domains were in register with the outer leaflet ordered domains. However, the induced inner leaflet domains appeared to be somewhat less ordered than those in the SM-containing outer leaflet, as judged by different partitioning of probes between Lo and Ld domains in the inner and outer leaflets. The basis of this conclusion is that a less ordered Lo domain has a weaker tendency to exclude probes that prefer localization in Ld relative to Lo domains (assuming all else being equal). This different partitioning was not the case for samples containing milk SM in place of egg SM. As in vesicles without cholesterol (see above), penetration of long milk SM acyl chains into the opposite leaflet (interdigitation) may have increased the extent of interleaflet coupling in those samples.

It was hypothesized that the induction of inner leaflet DOPC-containing ordered domains was aided by the migration of cholesterol out of the inner leaflet Ld domains and into the inner leaflet DOPC-containing ordered domains [46]. This was based on the observation that a dioleoyl probe lipid was depleted in the inner leaflet ordered lipid domains, and so DOPC should show similar depletion, which can only be compensated for by enrichment of cholesterol in those domains. Because a high cholesterol concentration increases membrane order, this cholesterol migration would aid the formation of the ordered inner leaflet domain. It is noteworthy that lateral cholesterol redistribution may explain why membranes with cholesterol rarely fail to show interleaflet coupling. However, lateral redistribution of cholesterol to support ordered domain formation should not occur if an outer leaflet was entirely composed of ordered state SM and cholesterol. That may explain why interleaflet coupling as measured by inner leaflet order appeared to be weaker in cholesterol-containing lipid vesicles in which SM almost completely replaced outer leaflet PC [15,16].

It should be noted that no domains were seen when egg SM was replaced by brain SM. However, it is likely that ordered submicroscopic nanodomains were present, as shown by later experiments using FRET, which detected nanodomain formation in asymmetric vesicles containing brain SM, DOPC and cholesterol (see below) [45].

Studies using the hemifusion method have extended these results [47]. Domain induction was studied in cholesterol-containing giant vesicles with one leaflet (in this case the inner leaflet) containing a mixture of SM (or the saturated lipid, DSPC) DOPC, and cholesterol, and the opposite leaflet containing DOPC and cholesterol. Again, ordered domains were induced in the DOPC leaflet lacking high-Tm lipids and were in register with those enriched in SM or DSPC in the inner leaflet. These outer leaflet DOPC domains were less ordered than those formed by the high-Tm lipids in the inner leaflet. In addition, it was found that the ordered domains in the high-Tm leaflet had somewhat decreased order relative to that in symmetric vesicles. Thus, interleaflet coupling in asymmetric membranes can alter the behavior of both leaflets relative to that in symmetric membranes when ordered domains are induced. It was also found that Ld domains (probably in both leaflets) became less ordered upon induction of ordered domains. The likely explanation proposed was migration of cholesterol from Ld domains to induced Lo domains, in agreement with the proposal from studies using cyclodextrin-exchange.

Interestingly, the SM forming large domains in the hemifusion studies was brain SM, for which large domains were not observed in the studies using cyclodextrin-exchange. The likely explanation is that the hemifusion studies used a lower concentration of cholesterol, and a lower cholesterol concentration can promote the formation of larger SM domains, at least in symmetric vesicles [48].

## 8. Experimental Studies in the Presence of Sterol: Suppression of Ordered Domain Formation Due to Interleaflet Coupling

Suppression of ordered domain formation in asymmetric membranes has attracted less attention. In addition to early examples in planar bilayers showing suppression of at least large size ordered domains, noted above, an important study in which cyclodextrin-catalyzed exchange was used to prepare asymmetric planar supported bilayers detected changes in domains upon loss of asymmetry [49]. In planar bilayers containing brain SM, DOPC and cholesterol in one leaflet and DOPC and cholesterol in the opposite leaflet, no domains were initially visible by microscopy. Upon decay of asymmetry, large domains formed. Induction of domains by loss of asymmetry was not distinguished from a transition from submicroscopic to microscopic domains.

The question of the effect of varying lipid structure upon induction or suppression of ordered domain formation in asymmetric large lipid vesicles prepared by cyclodextrin exchange was investigated in a FRET study in which saturated PC acyl chain length was varied [45]. Vesicles contained saturated PC, DOPC and cholesterol in the outer leaflet and DOPC plus cholesterol in the inner leaflet. Domain formation in asymmetric vesicles was compared to that in symmetric vesicles with a composition matching the outer leaflet composition of the asymmetric vesicles (ignoring possible cholesterol migration between leaflets after lipid exchange). For longer chain saturated PC (i.e., DSPC), symmetric and asymmetric vesicles had similar ordered domain thermal stability, i.e., similar Tm values. Although inner leaflet domain formation was not assayed, the behavior of giant vesicles with a similar lipid composition [47] indicates that the inner leaflet would contain induced ordered domains in the asymmetric vesicles. In other words, the outer leaflet was dominant, and likely to be so at all temperatures in which outer leaflet ordered domains were present (i.e., below their Tm).

As saturated PC acyl chain length was decreased, the behavior of asymmetric and symmetric vesicles diverged, with the inner leaflet becoming increasingly dominant in asymmetric vesicles as outer leaflet saturated PC acyl chain length was decreased. In other words, decreasing saturated PC acyl chain length increasingly allowed the inner leaflet DOPC to induce a disordered outer leaflet in the asymmetric vesicles even though the outer leaflet composition could still form ordered domains in symmetric vesicles. This showed up as a lower Tm for ordered domain formation in asymmetric vesicles relative to symmetric vesicles. For the saturated PC with the shortest acyl chain length studied (di C14:0 PC) no ordered domain formation could be detected in the asymmetric vesicles at any temperature (the lower temperature limit measured was about 15 °C), although ordered domains formed over a considerable temperature range in the symmetric vesicles. Intuitively, the effect of decreasing saturated PC acyl chain length in asymmetric vesicles is consistent with the decreased thermal stability of the ordered gel state in vesicles composed solely of saturated PC as acyl chain length is decreased (as judged by Tm values in pure lipid vesicles [31]).

Comparison of the behavior of symmetric and asymmetric vesicles containing the same total amount of saturated PC in the vesicles (as opposed to the same amount in their outer leaflets) showed that loss of asymmetry without a change in membrane lipid composition could induce ordered domain formation. In other words, the lower amount of high Tm lipid in the outer leaflet after asymmetry was lost, which destabilizes domain formation, was more than compensated by the stabilization of ordered domain formation when lipids with a strong tendency to form ordered domains were present in both leaflets. Because the assay involved FRET, this involved domain formation, not formation of large domains from preexisting nanodomains.

The effect of the amount of high-Tm lipid upon domain formation in symmetric vs, asymmetric vesicles was also examined [45]. This was tested using vesicles with brain SM, DOPC and cholesterol in their outer leaflet and DOPC and cholesterol in their inner leaflet. The asymmetric vesicles showed suppression of ordered domain formation at some (but not all) temperatures as judged by a lower Tm value in the asymmetric vesicles than that for symmetric vesicles with the same SM content as in the outer leaflet of the asymmetric vesicles. The degree of suppression increased as SM levels were decreased. This suggests that altering the level of SM could control whether ordered domains form in asymmetric cell membranes.

The observation that loss of asymmetry can trigger domain formation raises the interesting possibility that loss of asymmetry might be a trigger for raft formation in biological systems. However, the effect of loss of asymmetry noted above was detected for lipids (a short acyl chain saturated PC and DOPC) far from physiological. Therefore, a study was carried out in asymmetric vesicles using SM in place of saturated PC, and comparing DOPC to the more physiological POPC [50]. Vesicles contained SM, either unsaturated PC species, and cholesterol in the outer leaflet, and either unsaturated PC species and cholesterol in the inner leaflet. It was found that FRET-detected domain formation was suppressed by POPC relative to DOPC. This was true whether the POPC was the PC species in the inner or outer leaflet, and most pronounced with POPC as the PC in both leaflets. At least in the latter case, it was found that loss of asymmetry induced ordered domain formation. Thus, loss of asymmetry can induce ordered domain formation in a lipid mixture that is a reasonable first approximation for cell membranes.

The mechanisms behind the domain-suppressing effect of POPC (relative to DOPC) when in the same leaflet as SM (cis leaflet) and the domain-suppressing effect of POPC that is in the opposite leaflet as SM (trans leaflet) may not be the same. When in the same (cis) leaflet, the key property may be that POPC mixes more readily with SM than DOPC, i.e., the mutual miscibility of SM and POPC may be somewhat greater than for SM and DOPC. However, it should be noted that in symmetric vesicles the difference between POPC and DOPC in this regard seems small [34]. For the case of a SM-rich leaflet in contact with PC in the trans leaflet, the greater degree of coupling between leaflets when the trans leaflet has POPC relative to when the trans leaflet has DOPC, e.g., due to interdigitation, might increase the domain suppressing effect of trans leaflet POPC.

## 9. Effect of Differences between the Amount of Inner and Outer Leaflet Lipid upon Domain Formation

An issue that may influence which leaflet dominates domain forming behavior in asymmetric vesicles is the degree of lipid balance between the leaflets, i.e., the degree to which exchange does not change the ratio of lipid molecules in the outer leaflet to that in the inner leaflet, and the resulting differential stress under conditions of lipid imbalance [51,52,53]. (This is for lipids being exchanged in and out that normally occupy the same cross sectional surface area in a membrane. For exchanging in lipids normally taking up a different cross-sectional area in a membrane than the lipids being removed would mean that the number of lipids inserted by exchange weighted by the difference in cross sectional area vs. the number of lipids removed by exchange would determine if there was lipid balance.) Lipid exchange in asymmetric vesicles prepared by cyclodextrin-exchange appears to be close to 1:1, that is, one lipid molecule inserted for every one lipid molecule that is removed. This is based on the observation that vesicle size does not change substantially after lipid exchange [15,18]. Exchange may be close to 1:1 because the relative affinity of lipid for cyclodextrin vs. vesicles is dependent upon lipid balance. When vesicles have a deficit of lipid in the outer leaflet, lipid affinity for the vesicle relative to cyclodextrin would increase and lipid would tend to be deposited in the outer leaflet by cyclodextrin, while when vesicles have an excess of outer leaflet lipid and so the outer leaflet becomes crowded, extraction of outer leaflet lipid by cyclodextrin would become more favored. Alternately, or in addition, vesicles that develop an imbalance of lipid during exchange may not survive the exchange process, for example because they become leaky and release trapped sucrose needed for their isolation [16,18,54]. Even so, there could be a small but significant lipid imbalance between the inner and outer leaflets after lipid exchange, and this could impact domain formation. For example, an excess of lipid in the outer leaflet should favor the formation of a tightly packed ordered state, as ordered states are more tightly packed than disordered states. Likewise, a deficit of lipid would favor formation of a disordered state. Further studies are needed to investigate this issue. Comparing domain properties in asymmetric vesicles after exchange carried out at temperatures above and below Tm for a high-Tm lipid may result in different levels of imbalance and be useful in this regard. However, migration of cholesterol between leaflets in response to phospholipid imbalance may complicate interpretation.

## 10. Domain Formation in Plasma Membrane Preparations

The studies described above indicate that whether ordered domains are induced or suppressed in asymmetric vesicles, and domain physical properties in asymmetric vesicles, involve contributions from lipids in both leaflets, and that overall behavior depends heavily on the details of lipid structure and lipid composition in each leaflet. The ability to control lipid composition and structure may provide cells the ability to vary membrane organization and properties over a wide range. That this might be possible in vivo is hinted at by the number of situations in which lipid asymmetry is lost in vivo [5].

One direction for future studies investigating the biological role of lipid asymmetry is to investigate properties of ever more biologically realistic artificial asymmetric membranes. A second approach is to investigate the role of lipid structure and asymmetry in natural membranes. This is challenging, but two developments aid such studies. The first is the ability to prepare giant plasma membrane vesicles (gpmv) [55,56]. Despite the complexity of lipids in the plasma membrane, it has been shown that gpmv preparations can form large ordered domains visible by microscopy at low temperatures [57], and at physiological temperature can form ordered nanodomains detectable by FRET [58]. The second development is the ability to alter plasma membrane lipid composition by cyclodextrin-catalyzed lipid exchange [2,19]. This method can be used to almost totally replace outer leaflet phospholipids and sphingolipids with exogenous species, and can be used to prepare gpmv with altered lipid compositions and physical properties [2,19].

However, several issues complicate interpretation of gpmv behavior. First, formation of gpmv is associated with a loss of cytoskeletal interactions, which can perturb their behavior. Second it is not clear if the lipid composition of gpmv is exactly equal to that in the bulk plasma membrane. Third, domain formation is affected by the cell cycle, and thus likely to be affected by cell growth conditions. Fourth, the chemical reagents used to induce gpmv formation can modify membrane protein and lipid structure. Finally, there is at least a partial loss of lipid asymmetry when gpmv form and bud off of cells [59,60].

## 11. Association of Loss of Asymmetry with Ordered Domain Formation in Plasma Membranes

Recent work has clarified the relationship between gpmv domain formation and asymmetry [61]. Studies used gpmv prepared with N-ethyl maleimide (NEM), which reacts with free thiols, and should induce much fewer chemical modifications than the combination of dithiothreitol (DTT) and paraformaldehyde (PFA) more frequently used to prepare gpmv. Large domains and nanodomains were much less thermally stable in NEM gpmv preparations than in DTT/PFA gpmv [56,61]. Interestingly, NEM gpmv reacted with externally added annexin V much less than DTT/PFA gpmv, suggestive of a lower external PS concentration, and a greater level of asymmetry in NEM gpmv relative to that in DTT/PFA gpmv. Suppression of ordered domain formation in NEM gpmv due to asymmetry was detected by several experiments showing that treatments which destroy asymmetry result in stabilization of ordered domain formation in vesicles formed from gpmv lipids. This was observed whether or not membranes contained gpmv proteins. In addition, extensive digestion of gpmv proteins by proteinase K did not increase ordered domain stability. Although these experiments by no means rule out an important role of membrane proteins in domain formation, they do indicate that loss of lipid asymmetry is a major factor controlling ordered domain formation in gpmv. Whether this is also true for plasma membranes in intact cells is an important question for future studies.

## 12. Implications of Asymmetry Loss Due to Detergent Addition for Interpretation of Detergent Resistant Membranes Isolated from Cells

The observation that lipid asymmetry affects ordered domain formation may provide new insights into the origin of detergent resistant membranes from cells. Detergent resistant membranes can be isolated from living cells, and are rich in sphingolipids, cholesterol and proteins with a high affinity for ordered domains [62]. For this reason, in early studies detergent resistant membranes were considered an indicator that domains were present in living cells [62,63,64]. In addition, in symmetric artificial membranes detergent insolubility is strongly correlated with the presence of spectroscopically-detected ordered domains prior to detergent addition [65]. However, perturbations by detergent have been a concern, and a study in symmetric artificial membranes claimed to show that detergents could induce raft formation by promoting separation of lipids into different domains [66], although other studies in very similar symmetric artificial membranes show that detergents only increase the size of pre-existing domains [67]. A mechanism for induction of ordered domains not considered in prior studies is the impact of detergent on lipid asymmetry. Studies have shown that detergents can greatly increase the rate of lipid flip [68,69], and it is likely that addition of detergent to solubilize membranes significantly degrades lipid asymmetry. If that is the case, ordered domains could be induced by detergent addition under conditions that they are not present in the absence of detergent.

## 13. Conclusions

Studies in asymmetric vesicles that resemble plasma membranes in terms of their lipid asymmetry are providing a more realistic picture of the conditions in which ordered domains (rafts) form or fail to form. It is clear that the precise lipid composition of an asymmetric membrane is important to define under what conditions rafts are likely to form. Under some conditions the loss of asymmetry may be the trigger for raft formation in vivo. Further studies of both artificial and natural asymmetric membranes should help further clarify the rules for raft formation in vitro and in vivo.

## Figures and Tables

**Figure 1 membranes-12-00870-f001:**
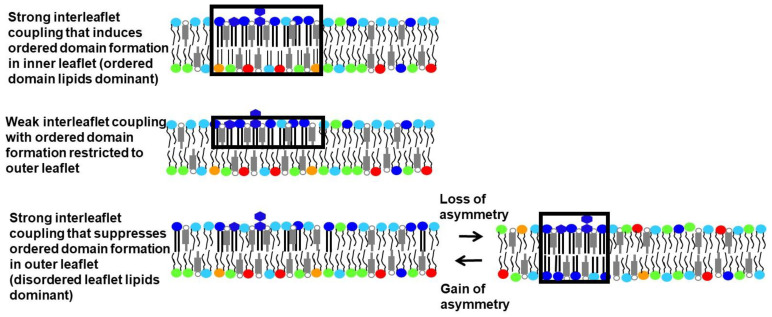
Some potential consequences of the loss or gain of lipid asymmetry for ordered domain formation. Ordered domains are shown by black boxes. Strong interleaflet coupling can either induce (**top**) or destroy (**bottom**) ordered domain formation in asymmetric membranes depending on whether the sphingolipid-rich or unsaturated lipid-rich leaflet dominates. In the latter case, loss of asymmetry (**bottom right**) can induce ordered domain formation. In the former case, loss of asymmetry can potentially inhibit domain formation (not shown). Dark blue, sphingolipid; light blue, PC; green, PE; red, PS; orange, PI; gray, cholesterol.

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
