# Peer review of "Ordered Domain (Raft) Formation in Asymmetric Vesicles and Its Induction upon Loss of Lipid Asymmetry in Artificial and Natural Membranes"

_membranes, 2022, doi:10.3390/membranes12090870_

Round 1

Reviewer 1 Report

Recent years have seen renewed interest in the compositional asymmetry of cell membranes, driven simultaneously by detailed lipidomics data, advances in experimental methods for preparing asymmetric model membranes, and theoretical work. The article by London reviews a subset of the literature of asymmetric membranes with a focus on interleaflet coupling of phase behavior, that is, how compositional asymmetry can induce or suppress domain formation. Dr. London’s group has made major contributions in this area starting with their development of robust methods for preparing asymmetric vesicles by cyclodextrin exchange, and the article nicely summarizes the present state of experimental knowledge (much of which originated in the London lab). Special emphasis is placed on the recent finding that in some cases, domain formation can be induced upon the loss of asymmetry. This has been observed not only in model membranes, but also in plasma membrane vesicles, suggesting a potential role for the controlled loss of asymmetry (e.g., through activation of scramblases) as a means of regulating lipid rafts in vivo.

This is a nice contribution and I have only a few minor suggestions:

1.     In the section titled “Stability of lipid asymmetry in artificial lipid vesicles”, readers may find it useful to know that addition of peptides can accelerate flip-flop (e.g., Doktorova et al. 2019 Biophys. J. 116:860, Nguyen et al. 2019 Langmuir 35:11735).

2.     Page 6 line 271: “It was also found that Ld domains become less ordered upon induction of ordered domains.” I assume that this means Ld domains in the outer leaflet (the one enriched in low-TM lipid) become less ordered? Some clarification would be helpful here.

3.     In the section titled “Effect of differences between the amount of inner and outer leaflet lipid upon domain formation”, it may be helpful to direct readers to recent theoretical work from the Deserno group about the influence of lipid number asymmetry on physical properties of bilayers including phase state (Hossein and Deserno 2020 Biophys. J. 118:624, Foley et al. 2022 Biophys. J. 121:2997, Varma and Deserno 2022 Biophys. J. https://doi.org/10.1016/j.bpj.2022.07.032).

Author Response

   For comments 1 and 3 we have added the requested references (and added some additional one).

   For comment 2 we have clarified that the decease in Ld order probably involved both leaflets.  

Reviewer 2 Report

The manuscript "Ordered domain (raft) formation in asymmetric vesicles and their induction upon loss of lipid asymmetry in artificial and natural membranes" by Erwin London is an interesting review on the actual topic of the science of lipids. It is well-written and well-organized, easy to read, which make this manuscript very interesting for the Membranes readers.

I would recomend to publish it after minor changes to improve the manuscript.

First of all, I believe the Review paper has to be generously illustrated with Figures and Schemes. For example, it will be very suitable to illustrate cyclodextrin-catalyzed exchange.

Secondly, I will suggest addding some Tables for more rapid interpretation of the text. E.g. in the Section "Association of loss of asymmetry with ordered domain formation in plasma membranes."

Thirdly, I would recomend to add one more section on the main experimental methods and techniques to study lipid domains and assymetry - it will help other researchers to set up the experimnents correctly and will increase the citation rate of the paper.

Author Response

   We have added references and directed the readers to more wide ranging recent reviews that have many additional figures and describe the various methods to prepare asymmetric membranes.   We do not think it is necessary to repeat that information in this much more narrowly focused review.